# The Perspective of Arctic–Alpine Species in Southernmost Localities: The Example of *Kalmia procumbens* in the Pyrenees and Carpathians

**DOI:** 10.3390/plants12193399

**Published:** 2023-09-26

**Authors:** Łukasz Walas, Marcin Pietras, Małgorzata Mazur, Ángel Romo, Lydia Tasenkevich, Yakiv Didukh, Adam Boratyński

**Affiliations:** 1Institute of Dendrology Polish Academy of Sciences, 62-035 Kórnik, Poland; lukaswalas@man.poznan.pl (Ł.W.); mpietras@man.poznan.pl (M.P.); 2Faculty of Biological Sciences, Kazimierz Wielki University, 85-064 Bydgoszcz, Poland; mazur@ukw.edu.pl; 3Botanical Institute of Spanish Research Council, 08038 Barcelona, Spain; angel.romo@gmail.com; 4Department of Botany, Ivan Franko National University of Lviv, 79005 Lviv, Ukraine; tasenkevich@gmail.com; 5M.G. Kholodny Institute of Botany, NAS of Ukraine, 01601 Kyiv, Ukraine; ya.didukh@gmail.com

**Keywords:** biogeography, climate change, ecological niche modeling, geographic range, relict plant

## Abstract

High-mountain and arctic plants are considered especially sensitive to climate change because of their close adaptation to the cold environment. *Kalmia procumbens*, a typical arctic–alpine species, reaches southernmost European localities in the Pyrenees and Carpathians. The aim of this study was the assessment and comparison of the current potential niche areas of *K. procumbens* in the Pyrenees and Carpathians and their possible reduction due to climate change, depending on the scenario. The realized niches of *K. procumbens* in the Pyrenees are compact, while those in the Carpathians are dispersed. In both mountain chains, the species occurs in the alpine and subalpine vegetation belts, going down to elevations of about 1500–1600 m, while the most elevated localities in the Pyrenees are at ca. 3000 m, about 500 m higher than those in the Carpathians. The localities of *K. procumbens* in the Carpathians have a more continental climate than those in the Pyrenees, with lower precipitation and temperatures but higher seasonality of temperature and precipitation. The species covered a larger area of geographic range during the Last Glacial Maximum, but its geographic range was reduced during the mid-Holocene. Due to climate warming, a reduction in the potential area of occurrence could be expected in 2100; this reduction is expected to be strong in the Carpathians and moderate in the Pyrenees.

## 1. Introduction

The origin of cold-adapted plant species and the formation of tundra in Northern Europe, Asia and North America took place in turn of the Pliocene/Pleistocene [1,2,3] as a reaction of the plant cover to the climate cooling. The plants adapted to the low temperatures and short vegetation periods in the Arctic zone and in the high mountains in southern regions evolved during approximately similar periods [3]. The Arctic plants reached mountains in Central Europe, Asia and North America, escaping to the south ahead of Pleistocene glaciers [1]. Inversely, the alpine plants could migrate to the north in the periglacial zone during deglaciations [4], as in the case of retreating glaciers, observed during the last few centuries in the Alps [5]. During interglacial periods of the Pleistocene (including the Holocene), the plant species connected with the cold climate could survive only in the Arctic and in the high mountains above the timberline, sometimes also on mires at lower altitudes [6,7,8], these being the glacial relicts [9,10].

High-mountain and arctic plant species are considered to be especially sensitive to climate change because of their specialized adaptation to the cold environment [11]. High temperatures and low humidity are expected to be the most important threats [12,13,14,15,16,17,18,19]. High risk also results from the forest line shift and expansion of trees, high shrubs and herbs, which could colonize or at least shade the sites of light-demanding tundra and alpine plant species [20,21,22,23,24]. 

*Kalmia procumbens* (L.) Gift & Kron & P.F.Stevens is an arctic–alpine, circumpolar, amphi-Atlantic plant [25,26,27]. It is an evergreen, dwarf, prostrate shrub, frequently creeping along the ground, especially at high elevations. In Northern Europe and in the mountains, it prefers the rocky ridges [24,28]. In Europe, it reaches its southernmost localities in the Pyrenees, Alps, northern Dinaric Alps and Carpathians. This species is one of the typical cold-adapted glacial relicts in the Central European mountains [29,30]. Despite this, it can survive in temperatures above 50 °C with a very high diurnal amplitude of temperatures [31].

In the Pyrenees and in the Carpathians, *K. procumbens* localities are confined predominantly to the alpine and sometimes to the subalpine vegetation belts. At lower locations of the subalpine belt, this species is connected mainly with the north-facing slopes [32,33,34], with relatively shorter vegetation periods, lower temperatures and temperature amplitudes, when compared with the south expositions [35,36,37,38]. It is stress-tolerant, adapted to extreme conditions of existence and characterized by slow growth in a relatively long growing season [39]. The formation of its photosynthetic apparatus is aimed at a minimum return of energy, which contributes to the conservation of resources. The population structure is dominated by plants of mature (generative) stages [40]. It prefers rocky ridges with small snow deposition during winter [28,38]. Such characteristics indicate that the species, although resistant to external influences, recovers rather slowly when it loses its position. The sites of *K. procumbens* are susceptible to influence of global climate change [28,38,41]. Despite that, recent observations indicated its more intense growth in the mountains of the Japanese islands [42] and even expansion in the Ukrainian Carpathians resulting from a reduction in snow cover [21]. The species had a broader potential ecological niche area at the global scale during the Last Glacial Maximum (LGM) than at present [41].

This study aimed to assess and compare the current potential niche areas of *K. procumbens* in the Pyrenees and Carpathians and their possible reduction due to climate change depending on the scenario. Additionally, its local, Carpathian and Pyrenean geographic ranges during the LGM were analyzed retrospectively based on the most probable high level of ecological niche conservatism in plants [43,44,45]. We expected the Carpathian populations would be more prone to the temperature rise and extension of the vegetation period than the Pyrenean ones. 

## 2. Results

### 2.1. Realized Geographic Niches

The distribution of *K. procumbens* in the Pyrenees and in the Carpathians is connected with the most elevated mountain massifs (Figure 1). The species has a more compact distribution of its localities in the Pyrenees than in the Carpathians. The altitudinal range in the Pyrenees is broader than that in the Carpathians. The altitudinal minima in both mountain systems are similar, but the most elevated localities in the Pyrenees have ascended about 400–500 m higher than those in the Carpathians (Figure 2). The current range of species differs from the potential range in the LGM (Appendix A), when the Carpathian arc was more suitable for the species, whereas conditions in the Pyrenees were less suitable.

The current potential niches of *K. procumbens* in the Pyrenees and Carpathians are determined first of all by elevation, whose influence on the current distribution of the species reaches nearly 80% in the Pyrenees and 60% in the Carpathians (Table 1). From the climatic variables, only precipitation of the driest month (bio 14) has a relatively high degree of influence in both mountain chains (Table 1). Additional important bioclimatic factors, with an influence on the potential niche of *K. procumbens* reaching 1% or more, are the temperature mean diurnal range (bio2), temperature annual range (bio7), and precipitation of the driest quarter (bio17) in the Pyrenees and the annual mean temperature (bio1), temperature seasonality (bio4), precipitation seasonality (bio15), and precipitation of warmest quarter (bio18) in the Carpathians (Table 1). 

The average values of every bioclimatic factor differed at a statistically significant level (*p* < 0.05) between localities of *K. procumbens* in the South and East Carpathians (the latter also includes one locality from the West Carpathians) (Table 2). The East Carpathian region of *K. procumbens* occurrence has generally higher precipitation (bio12–bio19) and is characterized by temperature factors (bio1, bio5, bio6, bio8–bio 11) that are slightly but statistically significantly higher than those in the South Carpathian region (Table 2). Populations from the Carpathians occur most often in the northeastern exposition, whereas in the Pyrenees, they occur in the northern exposition (Appendix A).

Principal component analysis (PCA) of the bioclimatic factors for the realized niches of *K. procumbens* indicated a separate grouping of Carpathian and Pyrenean localities. The East Carpathian localities are well separated from the South Carpathian ones, while Central Pyrenean localities are intermixed with East Pyrenean ones to a great extent (Figure 3). The estimation of potential ecological niches on the basis of the species localities in Central and East Pyrenees recognized the realized niches in both Pyrenean regions (Figure 4). Inversely, the estimation of the potential niches of *K. procumbens* in the East Carpathian localities recognized a realized niche in the East Carpathians, but not in the South Carpathians. Additionally, this combination indicated highly suitable conditions in the Tatra Mts. in the West Carpathians, where only one natural locality of the species currently exists. The estimation of potential ecological niches of *K. procumbens* using the South Carpathian localities did not detect suitable environmental conditions for the occurrence of this species either in the East or in the West Carpathians (Figure 4).

### 2.2. Future Geographic Niches

The potential niches highly suitable for *K. procumbens* in the Carpathians completely disappear in the year 2100, independently of the scenario of climate warming. Only a few South Carpathian populations could persist, mainly in the Fagarash, while the East Carpathian ones would not find suitable conditions (Figure 5). The potential niches in 2100 in the Carpathians would be determined by elevation with a contribution of ca. 50% in the East Carpathians and ca. 70% in the South Carpathians. The next restrictive factor in the East Carpathians would be precipitation of the warmest quarter (bio18), attaining a contribution of 30% or more (Table 3). The other bioclimatic factors influencing the potential niches in 2100 are the same as the current ones (compare Table 1 and Table 3), but the values are slightly smaller. 

In the Pyrenees, the situation of *K. procumbens* populations in 2100 would not be so drastically worse than that at present (Figure 6). The environmental conditions in the East Pyrenees allow the species populations to persist and even attain a broader area of potential niche distribution in the more optimistic scenario (Table 4). In the Central Pyrenees, the potential niche area suitable for *K. procumbens* in the high and very high levels would be restricted (Table 4). As in the present, the most influential factor would be elevation, reaching contributions of about 76 and 80% in the Central and East Pyrenees, respectively (Table 3). The remaining bioclimatic factors are the same as those determining the current ecological niches of the species in the Pyrenees (see Table 1 and Table 3).

The phytoindication method revealed a potential threat to *K. procumbens* occurrence in the East Carpathians within Ukraine. It results mainly from the changes in the hydrological regime of the species sites. The growth in average yearly temperature by 2 °C (pessimistic scenario) put *K. procumbens* at moderate risk of extinction, while that by 3 °C put *K. procumbens* at catastrophic risk of extinction (Figure 7).

## 3. Discussion

### 3.1. Realized Potential Niches in the Carpathians and Pyrenees

During the LGM, the potential range of *K. procumbens* was broader in the Carpathians, and for East Carpathians populations, the model also predicts suitable areas at a lower elevation, closer to the ice sheet. In the Pyrenees during the LGM, the potential range was located at lower elevations, and conditions for Central Pyrenees populations were almost unsuitable. In current conditions, this species attains its southernmost European localities in the Pyrenees and its close to southernmost European localities in the Carpathians [24,27]. *Kalmia procumbens* survived in both mountain chains in the subalpine and alpine vegetation belts due to the high-mountain climate with low temperatures and relatively high precipitation, mostly in the places with restricted snow cover during winters [21,22,28,38]. 

The low altitudinal borders of *K. procumbens* occurrence in the Carpathians and in the Pyrenees are at similar elevations, as a rule in habitats orographically or edaphically inaccessible for shrubs, tall herbs and grasses. The specific site conditions are mostly on the slopes exposed to the north, in rocky places with very thin layers of soil or on rocks completely without soil, and in places open to the wind. Sometimes, such conditions can be anthropogenic; for example, they could result from over-pasturing. This kind of pressure ceased over the last few decades and could be one of the reasons for the disappearance of the lowest localities of *K. procumbens* in the East Carpathians, as it was reported at 1455 m in the Chornokhora [46] but not found later [22,33]. The reduction in pastoralism in the Pyrenees during the last few decades could also cause the disappearance of the species’ lowest localities due to the expansion of the tall herbs and shrubs. 

The maximal altitudes of occurrence of *K. procumbens* in the Pyrenees are more elevated than those in the Carpathians. Such a rule was also observed in other subalpine and alpine plants that are common in the Pyrenees and Carpathians, such as *Juniperus communis* L. var. *saxatilis* Pall., *Salix reticulata* L., *Salix herbacea* L., *Salix hastata* L., *Dryas octopetala* L. and *Vaccinium gaultherioides* Bigelow (*V. uliginosum* L.) (Appendix A). The differences in the altitudinal maxima of the subalpine and alpine plants between the Carpathians and Pyrenees surely result from the higher elevations of the latter. The Carpathian arc is composed predominantly of medium-sized mountain ridges, with only three or four massifs being sufficiently well revealing, and several others with only a fragmentary developed alpine vegetation belt [47,48]. Inversely, in the Pyrenees, this type of vegetation is more frequent and covers a broader area [28,49,50,51]. 

*Kalmia procumbens* is well adapted to microhabitats with continental climatic conditions, to the extremely high daily amplitude of temperatures during the vegetation season [52,53] and early snowmelt [54], but the species could suffer from frost during the beginning of the vegetation season when development of the generative structures starts [55]. On the other hand, the late frost disturbances and high temperatures in the exposed places of *K. procumbens* occurrence have the effect of reducing most other plant species, promoting the successful regeneration of *Kalmia* [55]. In relation to the surrounding grass plant communities, for which the Index of Continentality is 19.7 or 16.5, after Gorchynsky and Rivas-Martinez, respectively, in *Loiseleurio-Cetrarietum* of the East Carpathians, it attains 21.6 and/or 17.5 [40]. Additionally, the average annual temperatures at the elevation of 1000 m in grass plant communities reach 2.6 °C, but in the rock coenoses dominated by *K. procumbens*, they reach 3.6 °C [19]. Nevertheless, *K. procumbens* does not reach its potential altitudinal maximum in most of the Carpathian ridges, as was determined for some other alpine shrubby plants in the East Carpathians [48]. 

### 3.2. Environmental Conditions of the Realized Niches of K. procumbens 

*Kalmia procumbens* is a calcifuge species occurring in mountains composed of metamorphic siliceous rock pH [28,56,57,58]. The plant communities with dominance of this species are classified as association *Cetrario nivalis-Loiseleurietum procumbentis* Br.-Bl. in Br.-Bl. and Jenny 1926 [59], from alliance *Rhododendro-Vaccinion* Br.-Bl. 1926. In the Pyrenees, the plant community *Cetrario nivalis-Loiseleurietum* is developed on the north-exposed, acid sites where the winds blow out snow, thus reducing snow cover [49,50,57,58,60]. A similar plant community is formed in the South Carpathians [61,62], and a fragmentary community is formed in the East Carpathians [38,56]. In the latter mountains and in the South Carpathians, *K. procumbens* occurs in the grassland communities on siliceous rocks [56,63,64]. 

The climate in the regions of *K. procumbens* occurrence in the mountains of Central Europe is of an oceanic to sub-continental type, cryo-oro-temperate thermotype and sub-humid to hyper-humid ombrotype [65,66]. Despite this, the average bioclimatic data retrieved from World Clim for *K. procumbens* localities in the Pyrenees revealed slightly milder conditions than those in the Carpathians (Table 2). The climate of the Carpathian localities of. *K. procumbens* appeared more continental with lower factors of temperature and precipitation and higher temperature seasonality. The continental climate of the steppes easterly and southerly from the Carpathians and at a closer distance to the continental climate of the central Euro-Asiatic continent could play some role in lowering positions of *K. procumbens* there, and the alpine and subalpine vegetation belt, in comparison with the Pyrenees.

The average bioclimatic factors of *K. procumbens* localities presented a low level of differences between the Central and Eastern Pyrenees, revealed mostly in the lower values of factors connected with precipitation in the central, more continental parts of this mountain chain (Figure 3). Nevertheless, the more humid climatic conditions in the Eastern Pyrenees can be a reason for the more dispersed and not-so-abundant localities of the species due to prolonged snow cover [28]. Similarly, the more Atlantic climate conditions in the Northern as opposed to the Southern Pyrenees [65.66] can explain a lower abundance and frequency of occurrence of *K. procumbens* on the southern macroslopes [28]. On the other hand, the higher and longer-lasting snow observed in the Eastern Pyrenees could reduce the potential habitats accessible for *K. procumbens*, as the species occurrence is connected mostly with places with thin snow deposits [21,22,28,38,54]. *Kalmia procumbens* occurs mostly on specific microhabitats, mainly the rocky ridges and rocks, where the snow is being blown away and the temperatures reveal high diurnal amplitudes. 

The bioclimatic differences between localities of *K. procumbens* in the East versus South Carpathians appeared higher than those between the East and Central Pyrenees (Figure 3). This finding could result from the higher elevations of the South Carpathian mountain massifs as opposed to those in the East Carpathians, and consequently, the greater number of *K. procumbens* populations reported from the higher elevations, which are characterized by lower temperatures and higher mean diurnal amplitude and isothermality.

The snow-free period appeared important for microsites inhabited by small ericaceous shrubs in the high mountains and in the arctic zone [67]. The longer snow-free period positively influenced the wood-ring increment in *Empetrum hermaphroditum* Hagerup [68], a species frequently occurring with *K. procumbens*. At the same time, less snowfall and a shorter duration of snow cover observed during the last decade were a reason for more abundant *K. procumbens* growth [21,69]. 

The differences in climatic conditions of the current localities of *K. procumbens* in the Pyrenees and Carpathians could result from (1) adaptation to different climates during the Holocene; (2) the origin of the Pyrenean and Carpathian populations from two different regions, namely the Arctic and the Atlantic for the Pyrenees and more continental for the Carpathians; or simultaneous action of both processes. 

### 3.3. Possible Influence of Climate Differences 

It is to be expected that the differences detected between the average climatic conditions of *K. procumbens* localities in the Carpathians and Pyrenees influenced the genetic structure of the species. The isolation of *K. procumbens* populations between the Pyrenees, Alps and Carpathians lasting at least during the Holocene [4,70,71] should constitute a reason for genetic and morphological differences. Gene exchange in Ericaceae is limited due to the low rate of seed dispersal [72] and restricted pollen transport, especially between populations from distant mountain chains [73]. These limitations explain the genetic and morphological differences, as described for other subalpine/alpine plant species, such as *Salix herbacea* L. [74], *Ranunculus glacialis* L. [75], *Saxifraga oppositifolia* L. [76], *Soldanella alpina* L. [77] and *Rhododendron ferrugineum* L. [78,79], and even provide a reason for speciation, as in the case of *Rhododendron ferrugineum* and *R. myrtifolim* Schott & Kotschy [80]. 

The worldwide genetic structure of *K. procumbens* detected using sequences of multiple nuclear loci revealed that southernmost European and East-Asiatic populations are genetically similar but different from the Arctic ones [81,82]. The southern populations of *K. procumbens* in the mountains diverged from the Arctic during the LG [82,83]. This is in contrast with the results of amplified fragment length polymorphism (AFLP) analyses, which indicated the isolation of the Central European mountain clade [84]. The number of verified populations and individuals used in these papers [82,84] was sufficient for the description of the general pattern of geographic differentiation of *K. procumbens* but was rather too small for its genetic differentiation in the Central European mountains. It could be expected that this species reveals differences between Pyrenean, Alpine and Carpathian localities; this hypothesis, however, should be verified in a specific study. 

### 3.4. Future Ecological Niches in the Carpathians and Pyrenees

A climate-change-caused reduction in the potential niches of *K. procumbens* at their southern limit of realized ecological niches in the mountains of Central Europe could be predicted and even expected, as in the case of other subalpine and alpine plants [10,28,29,38,41,48]. In that context, the drastic reduction in potential niches in the Carpathians in general, and the quite complete disappearance of suitable niches in the East Carpathians, is not surprising. The process of reduction in the geographic ranges of cold-adapted, high-mountain plants and their shifts to higher elevations is restricted in the first place by the highest mountain elevations. Extinctions could also result from rather slow uppermost colonization by the plants [85]. The East Carpathians do not possess massifs exceeding an elevation of 2500 m, which are the main centers of occurrence of alpine and subalpine plants [47,86]. It should be stressed that the process of the disappearance of the area of potential niches detected in the case of *K. procumbens* in fact concerns more or less entire alpine and subalpine flora in the East Carpathians [24,48]. 

In the Pyrenees, the reduction in potential niche area suitable for *K. procumbens* is less drastic than that in the Carpathians. This results from the more “alpine” character of the Pyrenees and the higher elevations of their highest peaks, which form the conditions for the occurrence of alpine flora [28,50]. The more intense reduction in potential niches suitable for *K. procumbens* in the Central as opposed to the East Pyrenees could be explained by the more continental climate of the Central Pyrenees, influenced by the close distance to the very dry and warm central part of the Ebro Basin from the south. This could reduce the influence of the Atlantic climate [58,66]. In fact, the westernmost localities of the species were detected on the northern macroslopes of the Pyrenees, with the more prominent impact of the Atlantic climate (Figure 1). This could also corroborate the amphi-Atlantic biogeographic character of *K. procumbens* proposed by Hultén [25]. 

The influence of the reduction in the area of potential niches to the currently realized niches of *K. procumbens* could be diminished by the presence of microrefugia suitable for the species within the area of its occurrence [87]. The species in fact settled in such microsites at their lowermost localities. The current localities of *K. procumbens* in the East Carpathians (Figure 1), however, exist in the area of a very low level of suitability of the potential niches, especially at low elevations, which are quite exclusively on the north-facing slopes. It could be expected that *K. procumbens* can exist for a long time outside the optimal environmental conditions on the northern slopes; this makes extinction following climate change more prolonged, especially for plants moderately sensitive to the lack of humidity [88] and heat [31]. Thus, the persistence of *K. procumbens* in environments with a very low level of suitability could result from the specific site conditions of the particular localities eliminating competition of the other plants, but it also could be caused by mycorrhiza. The symbiosis of *K. procumbens* with fungi [89,90] could mitigate the extinction rate, as plants with fungal symbiosis are less vulnerable to harsh environmental conditions [91,92]. 

The extinction rate of *K. procumbens* due to climate change could also be restricted by the shrub’s longevity, found to be more than 100 years [93]. This longevity with tolerance to harsh environments, particularly relatively high temperatures, high diurnal temperature amplitudes, winter frosts [24,52,53,94,95,96] and episodic summer frosts [97], has the effect of moderating the influence of climate change. *K. procumbens* plants covered with a shallow snow stratum are less vulnerable to spring frosts [98]. On the other hand, the speed of shifting of *K. procumbens* in the mountains after glacier regression could be rather moderate. In the Alps, this species exists together with other ericaceous shrubs, in areas with relatively stabilized plant cover, and colonizes new terrain about a century after glacier regression [5]. Our results suggest that cooler slopes may act as microrefugia, buffering the effects of increases in temperature on plant communities by delaying the extinctions of species with low temperature requirements [88].

The precipitation of the driest month (bio 14) and precipitation of the warmest quarter (bio 18) are the most influential bioclimatic factors for the present realized niche of *K. procumbens* in the East Carpathians in Ukraine. These bioclimatic factors act as limitations due to a lack of water in the vegetation period, moving *K. procumbens* out of the zone of acceptable risk when the average annual temperature rises by 2 °C and deep into the zone of catastrophic risk when the temperature rises by 3 °C. The role of other ecofactors is much lower and concerns only the acidity and salinity of soils, which are stable in the rocks in the localities of *K. procumbens*.

## 4. Materials and Methods

### 4.1. Study Areas

The Pyrenees and Carpathians were elevated during alpine orogenesis and currently conserve alpine flora with an abundance of endemic species [4,48,86,99,100]. The Pyrenees include several massifs reaching an elevation of more than 3000 m, with subalpine and alpine vegetation belts harbouring several arctic–alpine plants [28,50,51]. The Pyrenean range is divided into the western portion under Atlantic influence, the central portion with a more continental climate and the eastern portion under the influence of the Mediterranean climate [28,51]. *Kalmia procumbens* occupies elevated parts of the East and Central Pyrenees [28,34].

In comparison to the Pyrenees, the Carpathians cover a broader area and are more fragmented and divided into several mountain chains, but only a few of them are sufficiently high for the development of subalpine and alpine vegetation zones [47,48,86,101,102]. The localities of *K. procumbens* occupy the most elevated sites in the South and East Carpathians [33,103], with only one natural locality in the West Carpathians [104,105,106]. 

### 4.2. Data Sampling and Geographic Analyses

Data on the natural localities of the species were extracted from the Global Biodiversity Information Facility (GBIF) database, the literature, herbaria, and authors’ field notes. The geographic coordinates of localities were determined using Google Earth when not reported in the original data. In total, we gathered more than 2000 data, but after verification and exclusion of duplicates and questionable information, we analyzed 641 georeferenced data, 140 for the Carpathians and 501 for the Pyrenees. Maps of the distribution of *K. procumbens* in the Pyrenees and Carpathians were prepared using QGIS 3.16.4 “Hannover” [107]. The altitudinal ranges of the species in both mountain systems are presented in the graphs.

### 4.3. Environmental Variables

Temperature, precipitation and elevation determine the current (realized) ecological niches of species to the highest degree [108]. In spite of that, we used nineteen bioclimatic variables [109] and altitude (Table 1) to find factors that determine the current potential niche of *K. procumbens*. The usage of all these data could shed new light on the adaptation of the species to specific climatic data, retrieved from WorldClim (WC) database (http://worldclim.org/, accessed on 1 April 2023) [110]. For LGM (21 ka BP), we used data from PaleoClim (PC) (http://www.paleoclim.org/, accessed on 1 April 2023), which is based on the use of the CHELSA algorithm on PMIP3 data [111,112]. The spatial resolution of 30 arc-seconds (~1 km) of climate variables was applied. For the current climate (average for the years 1970–2000), we used WorldClim 2.1 database [110] bioclimatic data. For the future climate, we based our work on the Community Climate System Model (CCSM) [113] and used two representative concentration pathways (RCPs), RCP 2.6 and RCP 8.5 [114]. RCP 2.6 provides an increase in radiation forcing by 2.6 W/m^2^ and an increase in temperature by 1 °C before 2070 (average for 2061–2080), and RCP 8.5 provides an increase in radiation forcing by 8.5 W/m^2^ and an increase in temperature by 2 °C during the same period. Both are climate projections from GCMs that were downscaled and calibrated using WorldClim 1.4 as the baseline climate. 

PC used climate data from the CAPE project [115] and CCSM [113] for the delineation of potential niches during the Eemian Interglacial (LIG, 120–140 ka BP). These climatic data are based on geomorphological and geographical characteristics and do not take into account edaphic features and the structure of the substrate, which somewhat changes the microclimatic conditions. 

For retrospective analyses of the climate of the EM (LGM, 21 ka BP), the CHELSA algorithm was used on PMIP3 data. For the Mid-Holocene (MH) climate (ca. 6 ka BP), CCSM4 was used. For the current climate (average for the years 1970–2000), we used WorldClim 2.1 database [110] bioclimatic data. The previsions of future climate changes utilized were scenarios of two representative concentration pathways (RCPs), RCP 2.6 and RCP 8.5 [114]. 

The average values of bioclimatic variables were compared between the Carpathians and Pyrenees, as well as between regions within these mountain ranges. The Mann–Whitney U test conducted in the R environment was used for this purpose [116]. The influence of particular climatic variables on the current potential niches of *K. procumbens* in the Carpathians and in the Pyrenees was verified by principal component analysis (PCA). In PCA, the data for localities in the North Carpathians and South Carpathians and the data for localities in the Central and East Pyrenees were treated as separate groups. 

### 4.4. Niche Modeling

For the prediction of the potential range of *K. procumbens*, bioclimatic data related to its localities were used. MaxEnt 3.4.1. [117,118,119] was applied in analyses with maximum entropy modeling for the estimation of a probable distribution of the species outside its realized niche. The model with ENMeval R software [120] for the current climate was evaluated at first. The evaluation procedure followed that described by Salva-Catarineu et al. [121]. For the evaluation of the results of modeling, receiver operating characteristic (ROC) curves were used [122,123], and area under the curve (AUC) values below 0.6 were assessed as nearly random.

QGIS 3.16.4 “Hannover” [107] was applied for mapping the current and predicted potential niches on the climate variables. The potential distribution of *K. procumbens* was calculated for the different classes of suitability [45,121].

The phytoindication method was used for the assessment of econiches, indicators of the leading climatic and edaphic ecofactors, forecasting their changes depending on the climate for the Eastern Carpathians [40,124]. Modeling of econiche change and assessment of habitat loss threats for populations of the Eastern Carpathians were performed on the basis of the phytoindication data. For this purpose, the point values of the amplitude (x ± 2σ) were calculated for the leading ecofactors, and their changes were evaluated depending on the increase in average annual temperatures by 1, 2 and 3 °C. The acceptable risk zone is where the average values of the obtained data are inside the confidence intervals of ±2σ, and the catastrophic risk zone is where the amplitudes do not overlap, which means the complete disappearance of the species from this place [40].

## 5. Conclusions

*Kalmia procumbens* occurs in the alpine and subalpine vegetation belts, going down to elevations of about 1500–1600 m, while the most elevated localities in the Pyrenees are at ca. 3000 m, about 500 m higher than those in the Carpathians. The localities of *K. procumbens* in the Carpathians have a more continental climate than those in the Pyrenees, with lower precipitation and temperatures but higher seasonality of temperature and precipitation. Due to climate warming, a strong reduction in the potential area of occurrence by 2100 is expected in the Carpathians, and a moderate reduction is expected in the Pyrenees.

## Figures and Tables

**Figure 1 plants-12-03399-f001:**
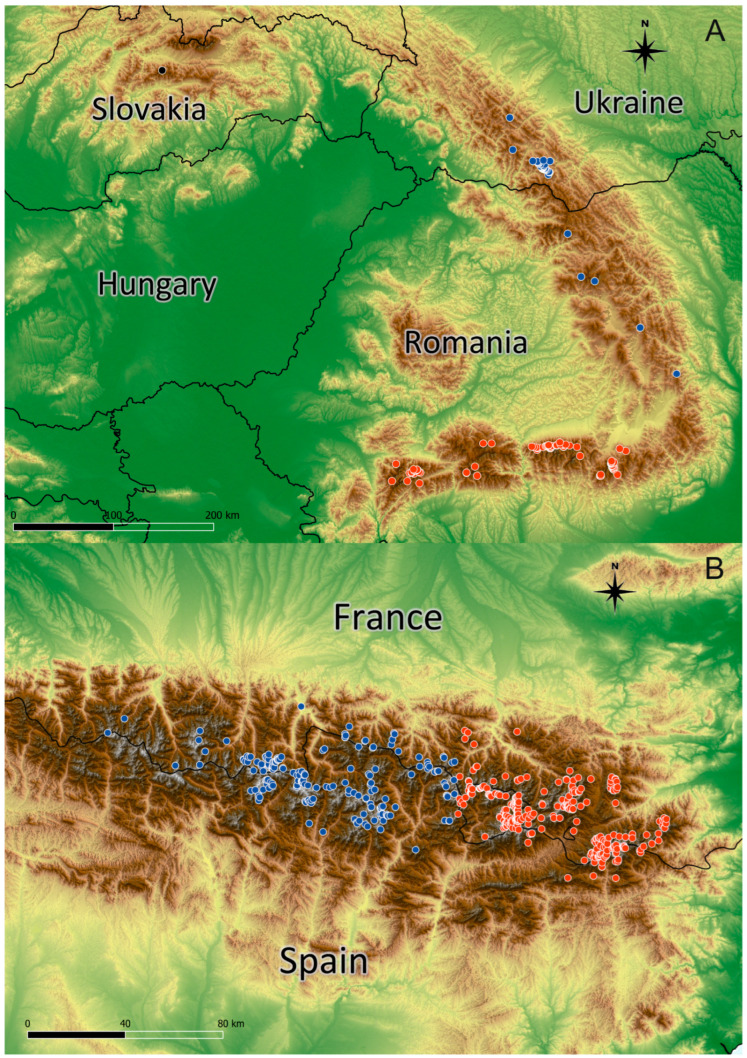
Geographical distribution of *Kalmia procumbens* on the basis of georeferenced data in the Carpathians (**A**) (blue dots—East Carpathians; red dots—South Carpathians) and in the Pyrenees (**B**) (blue dots—West Pyrenees; red dots—East Pyrenees).

**Figure 2 plants-12-03399-f002:**
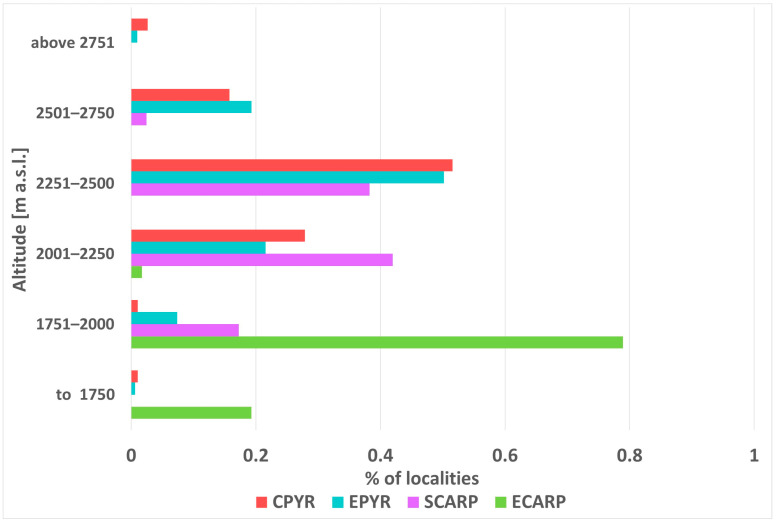
Vertical distribution of *Kalmia procumbens* localities in the Pyrenees (CPYR—Central Pyrenees; EPYR—East Pyrenees) and in the Carpathians (SCARP—South Carpathians; ECARP—East Carpathians).

**Figure 3 plants-12-03399-f003:**
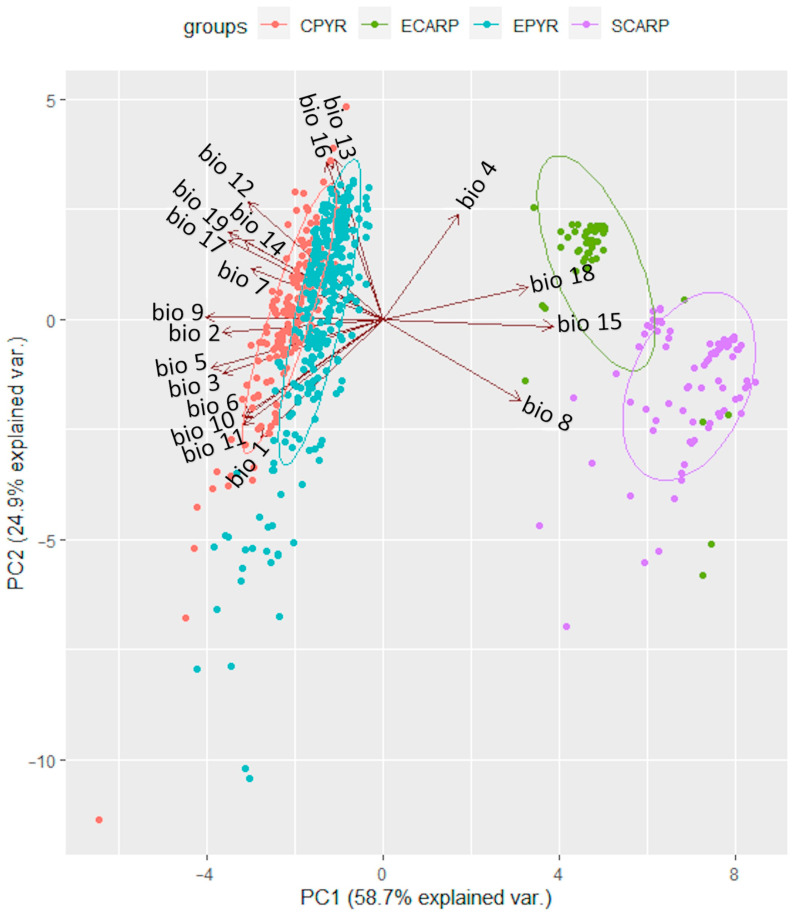
Position of localities of *Kalmia procumbens* from Central Pyrenees (CPYR), East Pyrenees (EPYR), East Carpathians (ECARP) and South Carpathians (SCARP) in PCA on the basis of bioclimatic variables (acronyms as in Table 1); ellipses indicate the 95% confidence intervals.

**Figure 4 plants-12-03399-f004:**
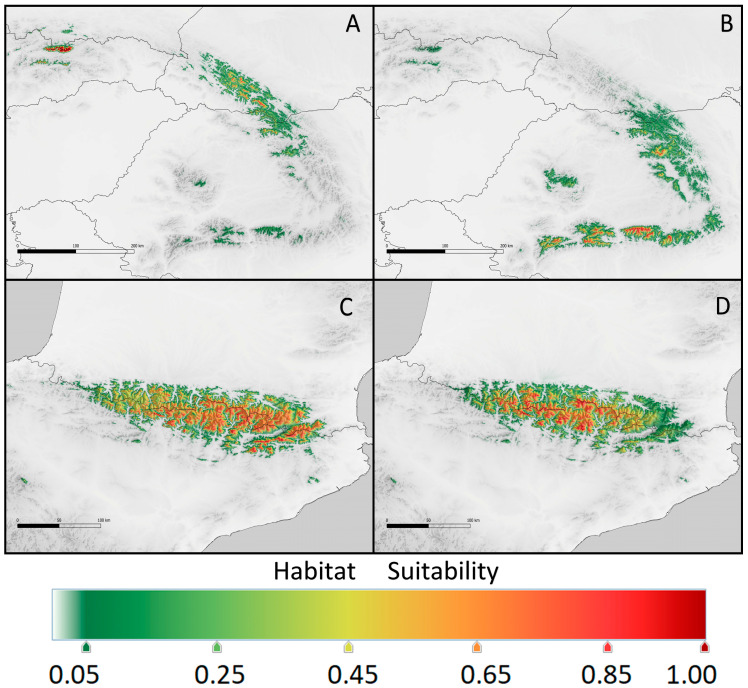
Current potential range of *Kalmia procumbens* in the Carpathians (**A**,**B**), estimated using environmental conditions from (**A**) the East Carpathians and (**B**) the South Carpathians, and in the Pyrenees (c and d), estimated using environmental conditions from (**C**) the East Pyrenees and (**D**) the Central Pyrenees.

**Figure 5 plants-12-03399-f005:**
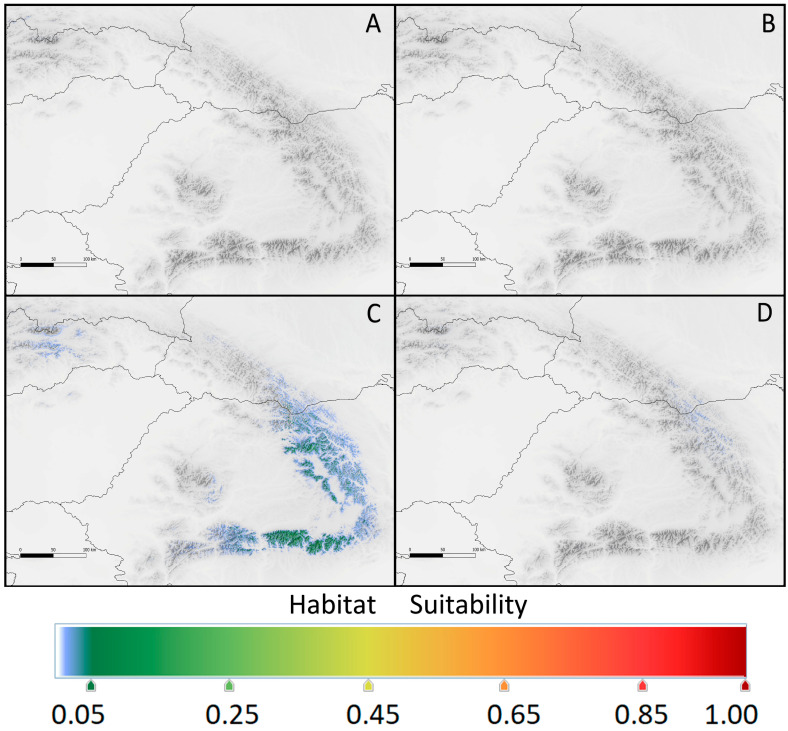
Provided potential range of *Kalmia procumbens* in the Carpathians in 2100 estimated using current environmental conditions from the East Carpathians for (**A**) scenario RCP 2.6 and (**B**) scenario RCP 8.5 and from the South Carpathians (**C**) for scenario RCP 2.6 and (**D**) scenario RCP 8.5.

**Figure 6 plants-12-03399-f006:**
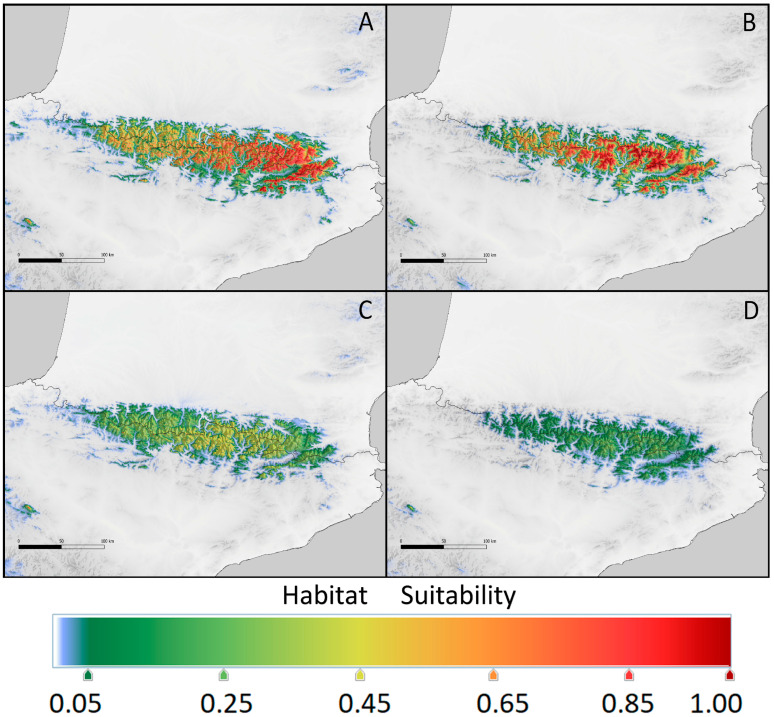
Provided potential range of *Kalmia procumbens* in the Pyrenees in 2100 estimated using current environmental conditions from the East Pyrenees for (**A**) scenario RCP 2.6 and (**B**) scenario RCP 8.5 and from the Central Pyrenees for (**C**) scenario RCP 2.6 and (**D**) scenario RCP 8.5.

**Figure 7 plants-12-03399-f007:**
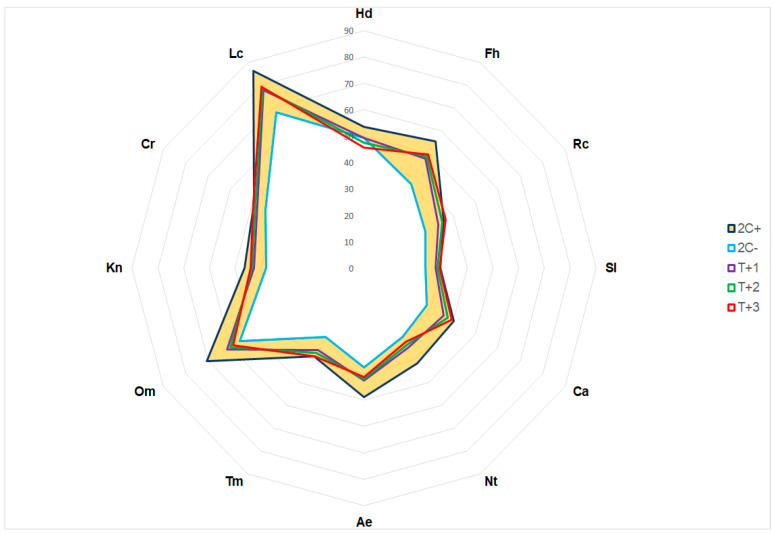
Percentage values of econiche factors of *Kalmia procumbens* characterization in the East Carpathians in Ukraine and potential changes depending on temperature changes: +1 °C (T + 1), +2 °C (T + 2) and +3 °C (T + 3); Hd—soil moisture, Fh—variability in soil moisture, Rc—soil acidity, Sl—soil salinity, Ca—soil carbonate, Nt—content of mineral nitrogen available for assimilation, Ae—soil aeration, Tm—thermal regime, Om—ombro regime, Kn—continental climate, Cr—cryo regime, Lc—illumination; indicator values 2C+ − X + 2σ, 2C− X − 2σ; the range of indicators x ± 2σ marked in orange.

**Table 1 plants-12-03399-t001:** Contribution (%) of bioclimatic variables and altitude to the realized habitats suitable for *Kalmia procumbens* in the Pyrenees (**PYR**), Central Pyrenees (CPYR), East Pyrenees (EPYR), Carpathians (**CARP**), East Carpathians (ECARP) and South Carpathians (SCARP); values of 1.0 and higher **bolded**.

	Bioclimatic Factor	CPYR	EPYR	PYR	ECARP	SCARP	CARP
	AUC	0.994	0.991	0.993	0.999	0.997	0.998
Bio1	Annual Mean Temperature	0.1	0.1	0.1	**2.0**	0.3	**1.3**
Bio2	Mean Diurnal Range	**4.1**	0.9	**2.5**	0.1	0.0	0.1
Bio3	Isothermality	0.6	0.2	0.4	**1.7**	0.0	0.9
Bio4	Temperature Seasonality	0.3	0.2	0.3	**4.7**	0.0	**2.4**
Bio5	Max Temperature of Warmest Month	0.7	0.3	0.5	0.1	0.1	0.1
Bio6	Min Temperature of Coldest Month	0.3	0.0	0.2	0.2	0.2	0.2
Bio7	Temperature Annual Range	**1.0**	**1.0**	**1.0**	0.1	0.0	0.1
Bio8	Mean Temperature of Wettest Quarter	0.4	**2.0**	**1.2**	0.2	0.2	0.2
Bio9	Mean Temperature of Driest Quarter	**3.8**	**6.0**	**4.9**	**5.5**	**8.2**	**6.9**
Bio10	Mean Temperature of Warmest Quarter	0.2	0.1	0.2	0.1	0.1	0.1
Bio11	Mean Temperature of Coldest Quarter	0.1	0.1	0.1	0.2	0.2	0.2
Bio12	Annual Precipitation	0.6	0.2	0.4	0.1	0.0	0.1
Bio13	Precipitation of Wettest Month	0.1	0.3	0.2	0.0	0.0	0.0
Bio14	Precipitation of Driest Month	**1.1**	0.4	0.8	0.0	0.1	0.1
Bio15	Precipitation Seasonality	0.3	0.1	0.2	**3.4**	**10.0**	**6.7**
Bio16	Precipitation of Wettest Quarter	0.3	**1.2**	0.8	0.0	0.0	0.0
Bio17	Precipitation of Driest Quarter	**7.8**	**6.2**	**7.0**	0.1	0.0	0.1
Bio18	Precipitation of Warmest Quarter	0.1	0.1	0.1	**31.1**	**9.9**	**20.5**
Bio19	Precipitation of Coldest Quarter	0.5	0.5	0.5	0.2	0.1	0.2
	Elevation	**77.6**	**80.2**	**78.9**	**50.1**	**70.4**	**60.3**

**Table 2 plants-12-03399-t002:** Average values of bioclimatic variables in the studied regions. According to the Mann–Whitney test, all differences between the Pyrenees and the Carpathians (bolded) are significant (*p* < 0.05), as are the differences between the Southern (SCARP) and Western (ECARP) Carpathians. For Eastern (EPYR) and Central (CPYR) Pyrenees, statistically significant differences (*p* < 0.05) were observed for bio2, bio3, bio5, bio7, bio12, bio13, bio14, bio15, bio16, bio17, bio18 and bio19 (shaded).

	Bioclimatic Factor	CPYR	EPYR	Pyrenees	ECARP	SCARP	Carpathians
bio1	Annual Mean Temperature	2.37	2.40	**2.39**	1.07	−0.25	**0.30**
bio2	Mean Diurnal Range	9.27	8.67	**8.90**	6.91	7.16	**7.06**
bio3	Isothermality	33.79	32.62	**33.07**	27.73	29.06	**28.52**
bio4	Temperature Seasonality	649.33	646.56	**647.61**	684.48	657.93	**668.81**
bio5	Max Temperature of Warmest Month	18.69	18.14	**18.35**	13.92	12.52	**13.10**
bio6	Min Temperature of Coldest Month	−8.73	−8.44	**−8.55**	−10.99	−12.08	**−11.62**
bio7	Temperature Annual Range	27.43	26.58	**26.90**	24.91	24.60	**24.73**
bio8	Mean Temperature of Wettest Quarter	−0.62	−0.30	**−0.42**	8.98	6.84	**7.73**
bio9	Mean Temperature of Driest Quarter	10.80	10.78	**10.79**	−6.81	−8.02	**−7.51**
bio10	Mean Temperature of Warmest Quarter	10.93	10.89	**10.90**	9.36	7.83	**8.46**
bio11	Mean Temperature of Coldest Quarter	−4.71	−4.60	**−4.64**	−7.19	−8.14	**−7.74**
bio12	Annual Precipitation	1434.57	1447.54	**1442.62**	1274.90	934.12	**1075.62**
bio13	Precipitation of Wettest Month	154.04	161.53	**158.69**	167.76	132.44	**147.06**
bio14	Precipitation of Driest Month	75.71	68.17	**71.03**	65.56	46.75	**54.58**
bio15	Precipitation Seasonality	20.19	23.53	**22.27**	32.36	39.56	**36.57**
bio16	Precipitation of Wettest Quarter	430.82	457.02	**447.08**	462.96	363.02	**404.52**
bio17	Precipitation of Driest Quarter	263.48	251.75	**256.20**	221.51	149.49	**179.33**
bio18	Precipitation of Warmest Quarter	263.57	251.91	**256.33**	462.67	358.05	**401.49**
bio19	Precipitation of Coldest Quarter	388.06	399.96	**395.45**	228.93	149.75	**182.63**

**Table 3 plants-12-03399-t003:** Contribution (%) of bioclimatic variables and altitude to models of the future potential range of *Kalmia procumbens* in the Central Pyrenees (CPYR), East Pyrenees (EPYR), East Carpathians (ECARP) and South Carpathians (SCARP); values of 1.0% and higher **bolded**.

	Bioclimatic Factor	CPYR		EPYR		ECARP		SCARP	
		RCP 2.6	RCP 8.5	RCP 2.6	RCP 8.5	RCP 2.6	RCP 8.5	RCP 2.6	RCP 8.5
	AUC	0.994	0.994	0.991	0.991	0.999	0.999	0.997	0.997
Bio1	Annual Mean Temperature	0.0	0.1	0.0	0.0	**1.5**	**1.1**	0.6	0.4
Bio2	Mean Diurnal Range	**3.9**	**3.1**	0.9	0.9	0.1	0.1	0.0	0.0
Bio3	Isothermality	0.8	**1.1**	**1.7**	0.1	**2.1**	**2.7**	0.0	0.0
Bio4	Temperature Seasonality	0.3	0.7	0.2	0.2	**4.2**	**4.9**	0.0	0.1
Bio5	Max Temperature of Warmest Month	0.8	0.3	0.2	0.5	0.1	0.1	0.1	0.1
Bio6	Min Temperature of Coldest Month	0.4	0.1	0.1	0.9	0.1	0.1	0.1	0.2
Bio7	Temperature Annual Range	0.9	0.5	0.4	**1.0**	0.1	0.1	0.0	0.0
Bio8	Mean Temperature of Wettest Quarter	**1.2**	0.7	0.4	**4.0**	0.0	0.1	0.1	0.2
Bio9	Mean Temperature of Driest Quarter	**4.3**	**5.5**	**6.4**	**4.0**	**9.5**	**3.8**	**6.6**	**5.9**
Bio10	Mean Temperature of Warmest Quarter	0.2	0.1	0.1	0.2	0.1	0.0	0.1	0.1
Bio11	Mean Temperature of Coldest Quarter	0.1	0.0	0.1	0.1	0.3	0.3	0.3	0.2
Bio12	Annual Precipitation	0.8	0.7	0.4	0.4	0.0	0.1	0.0	0.0
Bio13	Precipitation of Wettest Month	0.3	0.2	0.1	**1.1**	0.0	0.0	0.1	0.0
Bio14	Precipitation of Driest Month	**1.0**	**1.0**	0.4	0.5	0.0	0.1	0.0	0.1
Bio15	Precipitation Seasonality	0.3	0.3	0.0	0.1	**2.0**	**5.4**	**12.1**	**9.8**
Bio16	Precipitation of Wettest Quarter	0.5	0.3	0.2	0.3	0.0	0.0	0.0	0.0
Bio17	Precipitation of Driest Quarter	**7.5**	**8.7**	**7.3**	**5.5**	0.0	0.0	0.0	0.0
Bio18	Precipitation of Warmest Quarter	0.1	0.1	0.2	0.2	**32.0**	**30.3**	**9.6**	**10.2**
Bio19	Precipitation of Coldest Quarter	0.4	0.2	0.3	0.2	0.2	0.2	0.2	0.2
	Elevation	**76.1**	**76.4**	**80.5**	**79.9**	**47.5**	**50.5**	**70.0**	**72.4**

**Table 4 plants-12-03399-t004:** Area of potential range according to the tested model and four sets of stands: CPYR—Central Pyrenees, EPYR—Eastern Pyrenees, ECARP—Eastern Carpathians, SCARP—Southern Carpathians.

Region	Model	Area in the Probability Levels (Hectares)				Total
		Low (0.1–0.25)	Medium (0.25–0.50)	High (0.5–0.75)	Very High (>0.75)	
CPYR	Current	6099.48	6545.66	4920.93	1288.34	18,854.41
	RCP 2.6	6344.43	10,857.13	420.55	16.35	17,638.46
	RCP 8.5	6698.12	711.52	4.67	0.00	7414.31
EPYR	Current	4825.15	7280.36	8192.59	1338.57	21,636.67
	RCP 2.6	5756.46	5986.58	6773.90	5156.05	23,672.99
	RCP 8.5	4482.04	4812.61	4664.27	4470.36	18,429.28
ECARP	Current	7406.57	2796.66	1059.53	686.48	11,949.24
	RCP 2.6	0.00	0.00	0.00	0.00	0.00
	RCP 8.5	0.00	0.00	0.00	0.00	0.00
SCARP	Current	9808.72	4446.85	2123.17	708.15	17,086.89
	RCP 2.6	771.79	0.00	0.00	0.00	771.79
	RCP 8.5	0.00	0.00	0.00	0.00	0.00

## Data Availability

The data presented in this study are available from the corresponding author upon request.

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
