# Peer review of "The Perspective of Arctic–Alpine Species in Southernmost Localities: The Example of Kalmia procumbens in the Pyrenees and Carpathians"

_plants, 2023, doi:10.3390/plants12193399_

Round 1

Reviewer 1 Report

Dear authors

This study compares the ecological niche of Kalmia procumbens  in two mountain area. I think it is important to explain exactely the area that we used especially the differenc between western and eastern pyrenes ( administrative , border area).

My main concern is the GIBF Data. Indeed, Gibf Data contains data from many sources ( herbarium, observation , automatic identification); Furthermore,  what is the date of the different observation. Indeed, Gibf can contain old data and Climate change is already occurring that can mislead some information. I think a paragraph in the discussion as to be made to discuss the limit of Gibf data and to add information on the range of the data in the texte or supllementary

Author Response

This study compares the ecological niche of Kalmia procumbens in two mountain area. I think it is important to explain exactely the area that we used especially the differenc between western and eastern pyrenes (administrative, border area).

Response: We used geographic regions distinguished in the Carpathians and Pyrenees. The Carpathians subdivision follow those accepted by Mraz and Ronikier [47] and by Ronikier [86]. The regions of Pyrenees follows distinguished by Gómez et al. [28, 51], that is western under Atlantic influence, the central with more continental climate and eastern under influence of the Mediterranean climate.

The sentence was inserted in the current rows 390-392:

The Pyrenean range is divides into western under Atlantic influence, the central with more continental climate and eastern under influence of the Mediterranean climate [28, 51]

My main concern is the GIBF Data. Indeed, Gibf Data contains data from many sources (herbarium, observation, automatic identification); Furthermore,  what is the date of the different observation. Indeed, Gibf can contain old data and Climate change is already occurring that can mislead some information. I think a paragraph in the discussion as to be made to discuss the limit of Gibf data and to add information on the range of the data in the texte or supllementary

Response: Thank you very much for that comment. We used GBIF data with precaution and only those documented by the herbarium sources. The other could be doubtful, due to misidentification. We verified the main European herbaria, and we found some specimens determined as Kalmia/Loiseleuria procumbens represented Empetrum in fact. For this reason our materials were reviewed carefully and the data with any doubts were excluded, when do not confirmed in other sources. We used such a procedure as routine one, obligatory in every of our studies using geographic range modelling. Thank you also for the suggestion concerning the dates of the reports of particular localities. Following your comment we changed the sentence in the rows 406-410 (currently 401-406) to:

The geographic coordinates of localities were determined using Google Earth when not reported in the original data. Totally, we gathered more than 2000 data but after verification and exclusion of duplicates and questionable information we analysed 641 georeferenced data, 140 for the Carpathians and 501 for the Pyrenees.

Reviewer 2 Report

Dear authors

Lines 14-15: sensitive to climate changes because of close adaptations ....

Lines 25-27: during the Last Glacial, and reduced area during mid Holocene. Due to climate warming, a strong reduction of the potential area of occurrence to 2100 is expected in the Carpathians and a moderate reduction in the Pyrenees.

Lines33-34: The plants adapted to the low temperatures and short vegetation periods in the Arctic zone and in the high mountains in southern regions ...

Line 36: in central Europe,

Line 44: sensitive to climate changes ...

Lines 53-56: The species is one of the typical cold-adapted, glacial relicts in the Central European mountains [29,30]. Despite this, it can survive in temperatures above 50 °C with a very high diurnal amplitude of temperatures [31].

Line 72 and elsewhere: during the Last Glacial Maximum ...

Lines 87-88: The current range of species differs from the potential range ...

Line 96: which influences the current distribution ...

Lines 141-142: estimated in the East Carpathian localities recognized a realized niche ...

Lines 145-147: The potential ecological niche of K. procumbens estimated in the South Carpathian localities did not detect suitable environmental conditions for the species' occurrence either in the East or in the West Carpathians (Figure 4).

Line 160: attaining 30% or more (Table 3).

Lines 176-180: The environmental conditions in the East Pyrenees allow the persistence of the species populations and even attain a broader area of potential niche distribution in the more optimistic scenario (Table 4). In the Central Pyrenees, the potential niche area suitable for K. procumbens in the high and very high levels would be 179 restricted (Table 4).

Lines 189-191: The phytoindication method revealed a potential threat to K. procumbens occurrence in the East Carpathians in Ukraine. It results mainly from the changes in hydrological regime of the species sites.

Lines 206-207: the model predicted suitable areas also at the lower elevation, closer to the ice sheet. In the Pyrenees during LGM, the potential range

Lines 218-219: Sometimes such conditions can be anthropogenic, for example, could be affected by over-pasturing.

Line 227: for the Pyrenees and Carpathians, for example,

Line 233: and several others with only ...

Line 35: frequent and covers a broader area ...

Lines 238-239: could suffer from frost during the beginning of the vegetation season when ...

Lines 260-261: mountains of Central Europe ...

Line 275: can be a reason for the more dispersed and not-so-abundant localities...

Line 279: the higher and longer-lasting snow observed...

Line 285:  appeared higher than between East and Central Pyrenees

Lines 288-289: characterized by lower temperatures

Line 296: The differences in climatic conditions...

Line 298: the origin of the Pyrenean and Carpathian...

A thorough linguistic control by an English native speaker is needed as the quality of the language is not always the correct one.

Please always write the scientfic names in Italics.

A thorough linguistic control by an English native speaker is needed as the quality of the language is not always the correct one.

Author Response

Lines 14-15: sensitive to climate changes because of close adaptations ....

Response: the phrase was changed to:

sensitive to the climate change because of close adaptation

Lines 25-27 (currently 24-27): during the Last Glacial, and reduced area during mid Holocene. Due to climate warming, a strong reduction of the potential area of occurrence to 2100 is expected in the Carpathians and a moderate reduction in the Pyrenees.

Response: These phrases were changed to:

The species covered a larger area of geographic range during the Last Glacial, but reduced during the mid Holocene. Due to climate warming, a reduction of the potential area of occurrence could be expected in 2100, strong in the Carpathians, while moderate in the Pyrenees.

Lines33-34: The plants adapted to the low temperatures and short vegetation periods in the Arctic zone and in the high mountains in southern regions ...

Done

Line 36: in central Europe,

Done

Line 44: sensitive to climate changes ...

Done

Lines 53-56: The species is one of the typical cold-adapted, glacial relicts in the Central European mountains [29,30]. Despite this, it can survive in temperatures above 50 °C with a very high diurnal amplitude of temperatures [31].

Done

Line 72 and elsewhere: during the Last Glacial Maximum ...

Done

Lines 87-88: The current range of species differs from the potential range ...

Done

Line 96: which influences the current distribution ...

Done

Lines 141-142: estimated in the East Carpathian localities recognized a realized niche ...

Done

Lines 145-147:

The potential ecological niche of K. procumbens estimated in the South Carpathian localities did not detect suitable environmental conditions for the species' occurrence either in the East or in the West Carpathians (Figure 4).

Response: Your proposal would change the meaning of the sentence. Nevertheless we changed it inserting ‘using’ in place of ‘on’.

The potential ecological niche of K. procumbens estimated using the South Carpathian localities did not detect suitable environmental conditions for the species occurrence neither in the East, nor in the West Carpathians

Line 160: attaining 30% or more (Table 3).

Done

Lines 176-180:

The environmental conditions in the East Pyrenees allow the persistence of the species populations and even attain a broader area of potential niche distribution in the more optimistic scenario (Table 4). In the Central Pyrenees, the potential niche area suitable for K. procumbens in the high and very high levels would be restricted (Table 4).

Done

Lines 189-191:

The phytoindication method revealed a potential threat to K. procumbens occurrence in the East Carpathians in Ukraine. It results mainly from the changes in hydrological regime of the species sites.

Done

Lines 206-207:

the model predicted suitable areas also at the lower elevation, closer to the ice sheet. In the Pyrenees during LGM, potential range

Done

Lines 218-219:

Sometimes such conditions can be anthropogenic, for example, could be affected by over-pasturing.

Response: Sometimes such conditions can be anthropogenic, for example, could resulted from over-pasturing.

Line 227: for the Pyrenees and Carpathians, for example,

Done

Line 233: and several others with only ...

Done

Line 235: frequent and covers a broader area ...

Done

Lines 238-239:

could suffer from frost during the beginning of the vegetation season when ...

Done

Lines 260-261: mountains of Central Europe ...

Done

Line 275:

can be a reason for the more dispersed and not-so-abundant localities...

Done

Line 279:

the higher and longer-lasting snow observed...

Done

Line 285: 

appeared higher than between East and Central Pyrenees

Done

Lines 288-289: characterized by lower temperatures

Done

Line 296: The differences in climatic conditions...

Done

Line 298: the origin of the Pyrenean and Carpathian...

Done

Please always write the scientfic names in Italics.

Done

Comments on the Quality of English Language

A thorough linguistic control by an English native speaker is needed as the quality of the language is not always the correct one.

Response:

The English of the present version was corrected by Samuel Pyke.